# Exploration of a GMMA-Based Bivalent Vaccine Against *Klebsiella pneumoniae*

**DOI:** 10.3390/vaccines13030226

**Published:** 2025-02-24

**Authors:** Qikun Ou, Lu Lu, Lina Zhai, Shuli Sang, Yiyan Guan, Yuling Xiong, Chunjie Liu, Haibin Wang, Qiping Hu, Yanchun Wang

**Affiliations:** 1Department of Cell Biology and Genetics, School of Basic Medical Sciences, Guangxi Medical University, 22 Shuangyong Road, Nanning 530021, China; 18877961233@163.com; 2Laboratory of Advanced Biotechnology, Beijing Institute of Biotechnology, 20 Dongda Street, Beijing 100071, China; lulu52102022@163.com (L.L.); zhailina8023@163.com (L.Z.); sangshuli@bmi.ac.cn (S.S.); yiyiyiyan610@163.com (Y.G.); liucj@bmi.ac.cn (C.L.); 3Beijing International Science and Technology Cooperation Base for Antiviral Drugs, Beijing Key Laboratory of Environmental and Viral Oncology, College of Chemistry and Life Science, Beijing University of Technology, Beijing 100124, China; 4Department of Clinical Laboratory, The Fourth Medical Centre, Chinese PLA General Hospital, No. 51 Fucheng Road, Beijing 100037, China; yulingxx@163.com

**Keywords:** *Klebsiella pneumoniae*, outer membrane vesicle, generalized modules for membrane antigens, vaccine, immunization

## Abstract

Background: An emerging trend of mutual convergence between drug-resistant and highly virulent strains of *K. pneumoniae* has been identified, highlighting the urgent need for the development of novel vaccines. Methods: To delete the target genes and eliminate the plasmids carrying antibiotic resistance genes, CRISPR-Cas9 technology was employed to perform genome editing on a clinically isolated O2 serotype of *K. pneumoniae*. Subsequently, this strain was utilized as a host to express genes associated with the synthesis of O1 serotype LPSs to construct the recombinant strain capable of simultaneously expressing LPSs of both O1 and O2 serotypes. This recombinant strain was then used as the production strain for the preparation of vaccines based on GMMAs (Generalized Modules for Membrane Antigens), and its biological characteristics were characterized. Finally, the safety and immunogenicity of the vaccine were evaluated using mice as the model animals. Result: a GMMA vaccine characterized by a high yield and low toxicity was gained. Importantly, the lipopolysaccharides (LPSs) of both O1 and O2 serotypes of *K. pneumoniae* were successfully expressed on the surface of the outer membrane vesicles. Following immunization with the GMMA vaccine, mice were capable of producing antibodies against the GMMA and demonstrated resistance to the invasion of both serotypes of clinically isolated *K. pneumoniae*. Conclusions: The GMMA vaccine showed significant promise as a bivalent vaccine against *K. pneumoniae.*

## 1. Introduction

*Klebsiella pneumoniae* represents a frequently encountered Gram-negative bacterium in clinical settings and is capable of causing a diverse range of infections, including pneumonia, urinary tract infections, bacteremia, and liver abscess [1]. According to distinct evolutionary pathways, two principal clonal groups have progressively emerged. One of these groups displays multidrug resistance and is designated as classical *K. pneumoniae* (cKP) [2], while the other group is hypervirulent *K. pneumoniae* (hvKp), more virulent than cKp [3]. Recently, clinical doctors have been confronted with an even more formidable challenge: the fusion of the antimicrobial resistance determinants of cKp and the virulence factors of hvKp on the same or coexisting plasmids. hvKp, with multidrug-resistant (MDR) or extensively drug-resistant (XDR) characteristics, has been isolated from clinical samples [4]. Owing to the overuse and misuse of antibiotics, the incidence rate of multidrug-resistant (MDR) *K. pneumoniae* is on the increase [5]. Given that the development of antibiotics significantly lags behind the evolution of drug-resistant *K. pneumoniae*, the employment of vaccines to prevent *K. pneumoniae* infection represents an extremely promising approach.

The infection caused by *K. pneumoniae* is highly complex. Its isolates can be categorized into different serotypes and subsequently tracked via serotyping. Serotyping is founded on the recognition of specific antibodies in relation to diverse alterations in surface-exposed polysaccharides, namely the O antigen and the K antigen, thereby giving rise to a variety of O and K serotypes. The O antigen constitutes the outermost layer of the lipopolysaccharide (LPS), whereas the K antigen belongs to bacterial capsule polysaccharides (CPSs). It is estimated that there exist eight serotypes of the O-type antigen and seventy-seven serotypes of the K-type antigen [6]. Owing to the relatively smaller quantity of O antigens, currently, there is a tendency to utilize O antigens for vaccine preparation [7,8]. Notably, the proportion of O1 and O2 serotypes accounts for more than half of MDR *K. pneumoniae* [9]. Interestingly, the O1 antigen polysaccharide synthesis gene cluster contains just two additional genes, *wbbY* and *wbbZ*, compared to the O2 gene cluster [10,11]. This unique genetic characteristic renders it a potential target for a bivalent vaccine.

Currently, there are several types of vaccines for *K. pneumoniae* in the literature, including inactivated vaccines [12], ribosomal vaccines [13], protein vaccines [14], polysaccharide vaccines [15], conjugate vaccines [16], and outer membrane vesicle (OMV) vaccines [17,18,19]. Among them, the OMV vaccine is considered a promising vaccine due to its good immunogenicity and natural adjuvant properties [20]. The adjuvant activity of OMV is attributed to the presence of pathogen-associated molecular patterns (PAMPs) [21]. OMV contains proteins, lipids, nucleic acids, lipopolysaccharides (LPSs), and other substances that can activate the host’s innate immune defense mechanism and stimulate immune responses [22].

However, OMVs that are naturally secreted by bacteria are relatively scarce in quantity and possess natural forms of LPS, which might trigger systemic reactivity within the human body. Therefore, to address the issue of limited production and reduce endotoxin levels, it is necessary to genetically engineer bacterial strains to increase OMVs and decrease LPSs. We usually refer to these genetically modified OMVs as Generalized Modules for Membrane Antigens (GMMAs) [23]. GMMAs are considered a versatile platform for the design of effective multivalent combination vaccines [24]. Studies have shown that GMMA-based vaccines have alone given strongly immunogenic results and have not raised any concerns for animal health. At the same time, GMMAs are formulated without alhydrogel adjuvant, which would allow for further simplification of GMMA-based vaccine manufacturing [25]. The GMMA-based Shigella vaccine has entered the clinical research successfully [26].

In this study, to gain an ideal recombinant strain for GMMA vaccine preparation, the CRISPR-Cas9 system was employed to delete genes associated with LPS synthesis and OMV synthesis within a clinically isolated O2 serotype *K. pneumoniae* strain. This manipulation led to an augmentation in the production of OMV and a reduction in its toxicity. Subsequently, the O1 polysaccharide expression plasmid was introduced via electroporation, enabling the concurrent expression of both O1 and O2 polysaccharides on a single strain of *K. pneumoniae*, named KP-GMMA. When mice were immunized with the extracted GMMAs, it was observed that they could induce the generation of antibodies against both O1 and O2 serotypes. Additionally, they protected the mice against infection by the corresponding serotype of *K. pneumoniae*.

## 2. Materials and Methods

### 2.1. Bacterial Strains and Culture Conditions

All bacterial strains used in the present study are listed in Appendix A. Both the *K. pneumoniae* strain O1 serotype 304-1 (named as KP-O1) and O2 serotype 304-2 (named as KP-O2), taken from The Fourth Medical Center of the People’s Liberation Army General Hospital, were isolated from clinical samples. All *K. pneumoniae* derivatives were prepared in the laboratory and stored at −80 °C. The strains were warmed at 20–25 °C, resuscitated by the antibiotic-free LB (Luria–Bertani) solid culture plate streaking method, and placed at 37 °C. The next day, a single colony was inoculated into 5 mL of liquid LB medium and shaken at 220 rpm and 37 °C for 8–9 h to reach the plateau period before being applied to subsequent experiments.

The sequences of the genome and plasmids of the KP-O2 strain were obtained using Illumina Hiseq2000 platform and PacBio RS II sequencer at Majorbio Bio-Pharm Technology Co., Ltd. (Shanghai, China).

### 2.2. Recombinant Strain Construction

All plasmids used in the present study are listed in Appendix A. The primers used herein are listed in Appendix A. The knockout of the tolR gene may significantly enhance GMMA release by disrupting the cell membrane, whereas the modification of the Lipid A structure through the deletion of the lpxM and pagP genes leads to a diminished stimulation of the innate immune response by LPSs [27]. To construct the mutant strain for vaccine preparation, we employed the CRISPR-Cas9 gene-editing technology to knock out three specific genes, namely *tolR*, *pagP*, and *lpxM*, based on the clinically isolated *K. pneumoniae* 304-2 strain. A *K. pneumoniae* mutant strain was engineered using the method proposed by Anne-Catrin Uhlemann et al. [28]. The sgRNA site of the pUC19_CRISPR-DpmrA plasmid was modified into a pKP-CRISPR plasmid, which could be directly linked to the designed N20 sequence using a Seamless Assembly cloning kit (CloneSmarter, Cat.c5891-50), and the N20 sequence was designed using the online N20 design tool (https://crispor.gi.ucsc.edu/, accessed on 7 March 2023). For the recombinant template, the 500 bp sequence upstream and downstream of the target gene was selected and connected to the pKP-CRISPR plasmid through enzyme digestion and ligation. Then, 100–200 ng of the constructed PKP-CRISPR plasmid was electroporated into the strain, and the entire re-suspension was plated onto LB supplemented with zeocin at 1000 μg/mL. It was incubated at 37 °C overnight. After verifying successful plasmid electroporation through colony polymerase chain reaction (PCR), several clones were selected and inoculated in 5 mL LB supplemented with zeocin at 37 °C for two hours. Then, 100 μL of 10% arabinose was added, resulting in a final concentration of 0.2%. After induction at 30 °C and 220 rpm for 6–8 h, 100 μL culture was taken and applied to a zeocin-resistant plate for overnight cultivation. Finally, gene knockout was validated through PCR and DNA sequencing.

To eliminate the influence of drug-resistant plasmids, we employed the CRISPR-Cas9 system to remove two drug-resistant plasmids present in the strain, following previously published methods [29] and verification using the two distinct pairs of primers listed in Appendix A.

For the purpose of achieving the concurrent expression of both O1 and O2 polysaccharides, the O1 polysaccharide expression plasmid pACYC184-KPO1, which had been constructed by Liu Y et al. [10], was introduced into the O2 serotype mutant strain target strain via electroporation.

### 2.3. Production and Purification of GMMAs

An amount of 100 μL of the overnight cultured chassis strain was inoculated into 1 L of LB medium for overnight culture at 37 °C and 220 rpm. The resulting bacterial cells were centrifuged at 10,000× *g* for 15 min, and the supernatant was collected and filtered through a 0.45 μm membrane. The filtered solution was concentrated by ultrafiltration with a 100 KD ultrafiltration tube, ultracentrifuged at 32,600× *g* for 4 h at 4 °C, and the centrifuged OMVs were re-suspended with 2 mL of PBS, and then filtered through a 0.22 μm membrane.

### 2.4. Biological Characteristics Analysis of GMMAs

The morphology of GMMAs was observed using a transmission electron microscope (TEM). The average particle size, polydispersity index, and zeta potential of GMMAs were determined by the dynamic light scattering (DLS) method. Total protein content was quantified using the BCA protein assay. Protein composition was determined by 12% SDS-PAGE stained with Coomassie brilliant blue. (Genscript, Nanjin, China). Individual proteins in the GMMAs were determined using liquid chromatography–mass spectrometry.

LPS extracts were prepared using the Bacterial LPS extraction kit (Best Bio, Nanjing, China) following the manufacturer’s instructions. The quantification of LPSs took place using anthrone–sulfuric acid method. The characterization of LPSs was performed using a page gel Silver Staining Kit (Solarbio, Beijing, China).

To analyze the specific proteins in purified GMMAs or the expression of O1 serotype LPS, 10 μg of purified GMMAs or LPSs were separated on a 12% SDS-PAGE gel and transferred to a nitrocellulose membrane. Rabbit anti-*K. pneumoniae* O1 and O2 antibodies prepared by Zhu Li et al. [10] were used for co-incubation with the membrane at a dilution of 1:50. Finally, the bands were visualized by chemiluminescence after reaction with HRP-labeled anti-rabbit antibody.

The OMVs were enzymatically digested by employing the Filter-Aided Sample Preparation (FASP) technique. Subsequently, the protein expression levels were determined using an Orbitrap Fusion Lumos mass spectrometer, followed by data analysis using Proteome Discoverer.

### 2.5. Virulence Test of Galleria mellonella

We used the *Galleria mellonella* to compare the virulence of WT-KPand KP-GMMA strains. After overnight culture, the strains were diluted to 10^8^, 10^7^, and 10^6^ CFU/mL, and then 100 μL of bacterial solution was injected into the lower right foot of the *Galleria mellonella*. The survival rate was observed for 48 consecutive hours.

### 2.6. Immunization

Specific pathogen-free (SPF) BALB/c mice were purchased from Viton Lever Laboratory Animal Technology (Beijing, China), and housed at the Laboratory Animal Centre of the Academy of Military Medical Sciences. All animal experiments were approved by the Academy of Military Medical Sciences Animal Care and Use Committee (Approval Code: IACUC-DWZX-2023-022). The 6–8-week-old female BALB/c mice were immunized by subcutaneous injection (s.c.) on days 0, 7, and 14 of the trial. To explore the most appropriate immune dose, mice (n = 5/groups) received a subcutaneous injection of 1, 5, 10, and 20 μg of GMMAs or PBS according to the uniform immunization schedule. For evaluation of the immunoprotection, each group had 10 animals to test the corresponding indicators.

### 2.7. Biocompatibility Analysis

Three group mice were subcutaneously immunized with a 5 μg dose of GMMAs, WT-OMV, or PBS (used as negative control), and changes in weight and body temperature were monitored throughout the first week post-immunization. On the seventh day, once the mice’s weight had returned to baseline levels, the experiment was concluded, and the levels of biochemical markers such as alanine aminotransferase (ALT), aspartate aminotransferase (AST), alkaline phosphatase (ALP), lactate dehydrogenase (LDH), and blood urea nitrogen (UREA) were measured in the blood samples.

### 2.8. Analysis of Animal Infection and Inflammation

The bacterial challenge was performed 10 days after the last immunization (day 24). In the sepsis model, the control group and the vaccinated groups were intraperitoneally injected with sublethal doses of *K. pneumoniae* suspension (KP-O1 1 × 10^4^ CFU, KP-O2 1 × 10^7^ CFU), respectively. To analyze the bacterial load in different tissues, mice were sacrificed 48 h after the intranasal challenge with different serotypes of *K. pneumoniae*. The mice infected with bacteria were euthanized. Then, their livers, spleens, lungs, and kidneys were dissected and collected. A tissue grinder to was used to homogenize these tissues. After performing gradient dilution on the homogenate, 10 μL of the tissue dilution solution was pipetted and spread onto LB plates. Subsequently, the number of bacteria colonies on the plates was counted.

After dissection, the liver, spleen, lung, and kidney were removed and placed in 800 μL of sterile PBS. All the samples were mechanically homogenized and subjected to serial dilution. Diluted tissue homogenates were spread onto LB agar plates and incubated at 37 °C overnight. The bacterial numbers were counted the next day. To analyze the infiltration of inflammatory cells in tissues, tissue specimens were fixed with 4% paraformaldehyde fixative (Leagene, Beijing, China), processed according to the standard paraffin embedding process, and made into 4 μm sections for H&E staining.

### 2.9. Analysis of Animal Lung Functions

Following three immunizations as described in Section 2.6, mice (n = 5/group) were challenged by intraperitoneal injection with a different serotype of *K. pneumoniae*. After 48 h, mice were anesthetized intraperitoneally with 1% sodium pentobarbital and placed in an animal lung function analyzer (Anires2005, Beijing, China) to measure the lung function of mice, including the following parameters: Resistance of Lung (RL), Resistance of Expiration (RE), and Respiratory Dynamic Compliance (Cdyn).

### 2.10. Detection of Splenocyte Proliferation and Cellular Immune-Related Factors

One week after the final immunization, mouse spleens were aseptically removed and pressed through a fine nylon mesh using syringe plunges to prepare single-cell suspensions. These suspensions, at a concentration of 1 × 10^6^ cells per well, were cultured in plates containing RPMI 1640 medium supplemented with 10% (*v*/*v*) fetal bovine serum and 1% (*v*/*v*) penicillin-streptomycin (Thermo Fisher Scientific, Waltham, MA, USA). The spleen cells were stimulated with either GMMAs (2 μg/mL) or PBS for 72 h, respectively. During the final four hours of incubation, 20 µL CCK-8 solution was added. All measurements were performed at least in triplicate. Similarly, the supernatants derived from splenic lymphocytes that had been stimulated with GMMAs and PBS were harvested. Commercially available ELISA kits (Cusabio, Wuhan, China) were then utilized to quantitatively detect the levels of IFN-γ, IL-6, and IL-1β cytokines.

### 2.11. ELISA

In total, 100 μL of LPS or GMMA solution (100 μg/mL) was added to each well of a 96-well plate. After incubating at 4 °C overnight, the plate was washed three times with PBST. Then, 200 μL of ELISA blocking solution was added to each well and the plate was incubated at 37 °C for 2 h. After washing and drying, serially diluted serum (collected from immunized mice, 100 μL/well) was added to each well and incubated at 37 °C for 1 h. Then, the plate was washed three times and dried again. The horseradish peroxidase (HRP) conjugated anti-mouse IgG (Abcam, UK, 1:10,000 dilution) was added into each well (100 μL) and incubated at 37 °C for 1 h. After the reaction, the plate was washed and dried. A TMB Kit (CWBio, Suzhou, China) was used to initiate a color-producing reaction and measure the absorbance of each well at OD_450_.

### 2.12. Detection of Follicular Helper T Cells and Germinal Center B Cells

The inguinal lymph nodes of mice were placed on a 200-mesh sieve 3 days after the last immunization, and the sieve was rinsed with 1 mL PBS while grinding with a grinding rod to obtain a lymphocyte suspension. The supernatant was centrifuged (500× *g*, 5 min) and discarded and washed once with PBS. It was then blocked with Fc-block and stained with FITC-CD4, PE-PD-1, APC-CXCR5, APC-CD45, AF488-GL-7, and PE-CD95 antibodies. Detection took place using the Aurora Spectral Flow Cytometer (Cytek, Fremont, USA).

### 2.13. Statistical Analysis

All data were represented as mean ± standard deviation (SD) and analyzed using GraphPad Prism 9.5.0. Means were compared using a paired two-tailed *t*-test. The antibody levels were determined by one-way ANOVA followed by Bonferroni multiple pairwise comparison tests. Differences between treatment means were considered significant at *p* < 0.05.

## 3. Results

### 3.1. Construction and Evaluation of GMMA Chassis Strain of K. pneumoniae

Aiming to enhance the yield of OMVs and mitigate the toxicity associated with the natural forms of LPS, three specific genes of *K. pneumoniae* KP-O2 were knocked out sequentially. Identification through PCR using gene-specific primers (Appendix A) and DNA sequencing results showed that all target genes in the series were successfully knocked out (Figure 1A). At the same time, the plasmids that harbored resistance genes were also eliminated and validated by PCR (Figure 1B). To achieve the concurrent expression of both O1 and O2 polysaccharides, the O1 polysaccharide expression plasmid pACYC184-KPO1 was introduced into the mutant strain via electroporation. Finally, this strain was designated as the bead chassis strain KP-GMMA.

We examined the OMV proteins produced by wild-type *K. pneumoniae* 304-2 (WT-KP) and its modified chassis strain KP-GMMA and found that KP-GMMA had lower expression of high-molecular-weight proteins than WT-KP, while other major bands showed no significant changes (Figure 1C). To verify whether O1 and O2 lipopolysaccharides were expressed simultaneously, we extracted LPSs from wild-type *K. pneumoniae* (KP-O1-LPS, KP-O2-LPS) and KP-GMMA LPS (KP-GMMA-LPS). The characterization of LPSs using silver staining showed typical glycosylation step-like bands, and KP-GMMA-LPS produced a heterozygous polysaccharide between O1-LPS and O2-LPS (Figure 1D). Then, using the whole-cell antibodies against O1 serotype and O2 serotype pulmonary *K. pneumoniae* [10], the polysaccharide expression was verified again by Western Blot. We found that in the presence of antibiotics after electroporation of the pACYC184-KPO1 plasmid, the expression of O1-type LPSs covered the expression of natural O2-type LPSs, and only heterozygous LPSs could be expressed in the absence of antibiotics (Figure 1E,F).

### 3.2. Characterization of GMMAs

After culturing the bacteria, OMVs from wild-type *K. pneumoniae* (WT-OMV) and GMMAs were isolated via ultracentrifugation. Transmission electron microscopy (TEM) was subsequently employed to examine these vesicles, revealing irregular spherical structures characterized by lipid bilayers (Figure 2A,B). DLS analysis determined the average particle size to be approximately 120 nm (Figure 2C). The stability of the GMMA vaccine was then assessed, demonstrating no significant changes when stored at room temperature (Figure 2D–F). The analysis of GMMA yield indicated that mutant strains produced approximately three times the amount of GMMAs compared to the wild strain (Figure 2G), while the LPS content was reduced (Figure 2H). To visually assess toxicity, *Galleria mellonella* larvae were used, revealing higher survival rates when challenged with the mutant strain compared to the wild-type strain under various conditions (Figure 2I). To examine the alterations in OMV proteins following mutation, we employed mass spectrometry analysis to compare the expression levels of the top ten proteins (see Appendix A).

### 3.3. Immune Dosage and Biocompatibility of GMMA Vaccine

Additionally, to evaluate the potential of GMMAs as a vaccine candidate against bacterial infections, we investigated organ damage in mice subjected to different immunization regimens (refer to Appendix A). The findings demonstrated a significant difference in IgG antibody titers between groups receiving three immunizations and those receiving two. Consequently, a three-immunization regimen was selected. We observed that higher immunization doses caused slight damage to the renal tubules of mice (Appendix A). After a thorough evaluation of both immunogenicity and safety profiles, an optimal immunization dosage of 5 μg was established. We observed that higher immunization doses caused slight damage to the renal tubules of mice. Based on these findings, an optimal immunization dosage of 5 μg was established. Biocompatibility is a critical parameter in the assessment of vaccines. Changes in body temperature and the weight of immunized mice were monitored over the course of the first-week post-immunization (Figure 3A). Compared with the WT-OMV group, the body weight of mice vaccinated with GMMAs returned to the pre-immunization level earlier (refer to Figure 3B) and there was no significant difference in the body temperature changes of the three groups of mice (refer to Figure 3C). For the blood ALT, AST, ALP, LDH, and UREA levels, there were no significant differences among the three groups (refer to Figure 3D). Subsequently, we also carried out histological analysis on the liver, lung, spleen, and kidney tissues using Hematoxylin and Eosin staining (H&E) (Figure 3E). No significant pathological damage was observed in the lungs, spleens, or livers after three immunizations in all groups. All these results suggested that the GMMA vaccine exhibits good biocompatibility.

### 3.4. The GMMA Vaccine Is Capable of Eliciting Both Humoral and Cellular Immune Responses

Mice were subjected to subcutaneous immunization with 5 μg of GMMA on days 0, 7, and 14. Subsequent analyses were conducted to evaluate the humoral and cellular immune responses induced by GMMAs (Figure 4A). Finally, the serum IgG antibody titers in the immunized mice were quantified using ELISA (Figure 4B). To assess the polarization of the immune response towards a Th1 or Th2 profile, we analyzed the antibody subtypes IgG2a and IgG1 and calculated their respective ratios. As depicted in Figure 4E, the IgG2a-to-IgG1 ratio was approximately 1, suggesting a balanced systemic IgG1/IgG2a response induced by the GMMA vaccine, which implies the simultaneous activation of both Th1 and Th2 immune responses (Figure 4C). Furthermore, splenic lymphocytes from the mice were harvested and stimulated with GMMAs as an antigen, promoting the proliferation of antigen-specific lymphocytes (Figure 4D). Concurrently, the supernatant from the stimulated splenic lymphocytes was collected to measure cytokine secretion levels, including for IL-6, IL-1β, and IFN-γ. The results demonstrated that cytokine levels were significantly elevated compared to the control group (Figure 4E). To facilitate a more comprehensive understanding of immune cell activation and proliferation, the inguinal lymph nodes from immunized mice were collected. Flow cytometry was subsequently utilized to assess the proportions of T follicular helper (Tfh) cells and germinal center B (GC B) cells both pre- and post-immunization. Notably, the quantity of Tfh cells increased fivefold compared to pre-immunization levels (Figure 4F), while the number of GC B cells exhibited a tenfold increase (Figure 4G). These findings support the conclusion that the GMMA vaccine is capable of inducing both humoral and cellular immune responses.

### 3.5. GMMA Vaccine Can Protect Mice from Diverse Serotype K. pneumoniae Infection

To evaluate the immunoprotective efficacy of GMMAs against various serotypes of *K. pneumoniae*, we initially endeavored to investigate their protective capacity against the O1 serotype of *K. pneumoniae* 304-1 (KP-O1) (Figure 5A). Following the final immunization with the GMMA vaccine, Western Blot (WB) analysis was carried out on the total bacterial protein of KP-O1 using mouse serum (Figure 5B). This demonstrated that the immunized serum could specifically recognize the protein of the KP-O1 bacterial strain. Ten days after the last immunization, an intraperitoneal challenge was administered with a non-lethal dose of 1 × 10^4^ CFU KP-O1, and the lung function of mice was assessed three days later. It was observed that respiratory resistance increased in the control group (Figure 5C), indicating that these mice experienced breathing difficulties following infection by *K. pneumoniae*. Concurrently, lung compliance decreased, and lung volume contracted, showing significant differences compared to the vaccine group. This suggests that the GMMA vaccine can alleviate lung inflammation and provide a protective effect on the lungs of mice. To assess the GMMA vaccine’s efficacy against *K. pneumoniae*-induced bacteremia, we evaluated bacterial loads in mouse organs (Figure 5D). Unlike the control group, the vaccine group showed minimal detectable bacteria, indicating the vaccine’s ability to limit local infections and prevent systemic spread. Pathological examination of tissue sections (Figure 5E) revealed significant bleeding in the PBS group, while the GMMA group showed no major pathological changes. These findings suggest that the GMMA vaccine effectively prevents infection by the O1 serotype of *K. pneumoniae*.

To demonstrate the protective efficacy of the GMMA vaccine against various serotypes of *K. pneumoniae*, we conducted a challenge experiment using *K. pneumoniae* 304-2 (KP-O2) of the O2 serotype at a concentration of1 × 10^7^ CFU (Figure 6A). Consistent with previous methodologies, we assessed the specific interaction between the serum and O2 serotype bacteria (Figure 6B), as well as lung function (Figure 6C) and bacterial load (Figure 6D). The PBS-treated group exhibited respiratory obstruction and symptoms of bacteremia, whereas the GMMA vaccine group showed no significant deviation from normal mice. The bacterial load in the organs was nearly undetectable, as confirmed by HE staining (Figure 6E). A summary of a brief immune protection experiment is shown in Table 1. In conclusion, the GMMA vaccine effectively confers protection against both O1 and O2 serotypes of *K. pneumoniae*.

### 3.6. Nucleotide Sequence Accession Number

The complete genome sequence of *K. pneumoniae* KP-O2 has been deposited in GenBank under the accession numbers CP178192 (Chromosome), CP178193 (plasmid A), CP178194 (plasmid B), CP178195 (plasmid C), CP178196 (plasmid D), and CP178197 (plasmid E).

## 4. Discussion

*K. pneumoniae* has consistently been identified as one of the most prevalent hospital-acquired pathogens worldwide. It is also a significant contributor to neonatal sepsis, ranking among the top three causative agents in the majority of cases [30,31,32,33]. In addition to the high frequency of *K. pneumoniae* infections, the widespread phenomenon of antibiotic resistance presents a substantial challenge [34]. Recently, evolutionary processes have led to the emergence of MDR and XDR strains within the *Enterobacteriaceae* family, which exhibit resistance to nearly all currently available antibiotics [35,36].

Vaccination is a vital strategy for the prevention of infectious diseases [37]. In addition to protein and capsular polysaccharide (CPS, K-antigen) vaccines, LPS-based vaccines represent a major focus in *K. pneumoniae* vaccine research. Both purified natural and semi-synthetic LPSs have been tested as vaccine candidates [38,39]. OMV vaccines demonstrate advantages by combining the immunogenicity of both polysaccharides and proteins while avoiding the safety risks associated with attenuated/inactivated vaccines. Augmented by the immune-stimulating effects of Toll-like receptor (TLR) agonists, the OMV vaccine possesses inherent adjuvant properties. These agonists, known as PAMPs, activate pattern recognition receptors (PRRs), thereby making OMVs a powerful catalyst for innate immune responses [40]. Current OMV vaccines targeting *K. pneumoniae* are derived either from natural OMVs [17,18] or from bacterial biomimetic vesicles produced under high mechanical pressure [19]. Currently, research on OMV-based *K. pneumoniae* vaccines is not enough [17,18,19]. Although these studies have shown promising protective efficacy, none have resolved the potential toxicity caused by the natural LPS in native OMV vaccines. In this work, we pioneered the application of GMMA technology in *K. pneumoniae* vaccine research. Compared to native OMVs, GMMA-based vaccines demonstrate enhanced safety profiles while retaining the capacity to carry LPSs of diverse serotypes, highlighting their potential as multivalent vaccine candidates. Developing such novel multivalent formulations is a prioritized objective in advancing *K. pneumoniae* vaccine research. In our study, we employed CRISPR-Cas9 to construct a triple-mutant strain (*tolR*, *pagP*, and *lpxM*) addressing both vesicle yield and LPS toxicity. Experimental validation demonstrated a threefold increase in GMMA production compared to wild-type OMVs, with significantly attenuated toxicity—a process termed GMMA detoxification. This modification primarily reduces lipid A acylation and phosphorylation, thereby diminishing Toll-like receptor 4 (TLR4) activation capacity [41,42,43,44]. Furthermore, recent studies have demonstrated that OMVs possess the potential to act as vectors for the transmission of bacterial resistance genes [45,46]. In response to this finding, we proceeded to knock out three specific resistance genes, namely blaKPC-2, located on plasmids B and E, blaCTX-M-15, and blaTEM-1B, using targeted genetic manipulation techniques. This strategic intervention aimed to reduce the risk of antibiotic resistance transfer and improve the overall safety profile of the system.

While GMMAs are generally recognized for their versatile “plug and play” characteristics, our primary focus is on their ability to retain and present secreted vaccine antigens on their surface. Through a comprehensive analysis of the genetic differences between the O1 and O2 serotype polysaccharides of *K. pneumoniae*—where the gene for the O1 serotype polysaccharide contains an additional pair of *wbbY* and *wbbZ* gene clusters compared to the O2 serotype—we incorporated these two gene clusters into the KP-O1 plasmid via electroporation. This genetic modification facilitated the simultaneous expression of both O1 and O2 LPS on GMMAs. However, our results indicated the production of a heterozygous polysaccharide.

The mechanism underlying polysaccharide synthesis requires further in-depth analysis in future studies. Subsequent experiments have conclusively demonstrated that immunized mice can generate antibodies against both O1 and O2 LPS simultaneously, effectively defending against two distinct serotypes of *K. pneumoniae*. As a result, a bivalent vaccine against *K. pneumoniae* has been successfully developed. Research findings indicate that multiple antigens can coexist on a single GMMA particle without causing immune interference. Consequently, our future plans involve the continuous expression of all remaining O antigens on GMMAs to develop multivalent vaccines.

However, our research is not without limitations. The heterogeneity in the size of the OMVs presents a significant challenge in vaccine quality control. Utilizing hydrogels or gold nanoparticles for encapsulation may offer a viable solution to address this issue. Furthermore, a systematic evaluation of the long-term effects of immune protection is lacking. Future research should focus on obtaining specific experimental data to inform the development of GMMA vaccines.

## 5. Conclusions

In conclusion, through the genetic modification of clinically isolated *K. pneumoniae*, a GMMA preparation strain was developed, leading to the construction of a GMMA-based *K. pneumoniae* vaccine. It was observed that this vaccine successfully induced the production of antibodies targeting both the O1 and O2 serotypes and protected the mice against infections caused by two serotypes of *K. pneumoniae*. This study offers a novel approach to the development of a bivalent *K. pneumoniae vaccine*.

## Figures and Tables

**Figure 1 vaccines-13-00226-f001:**
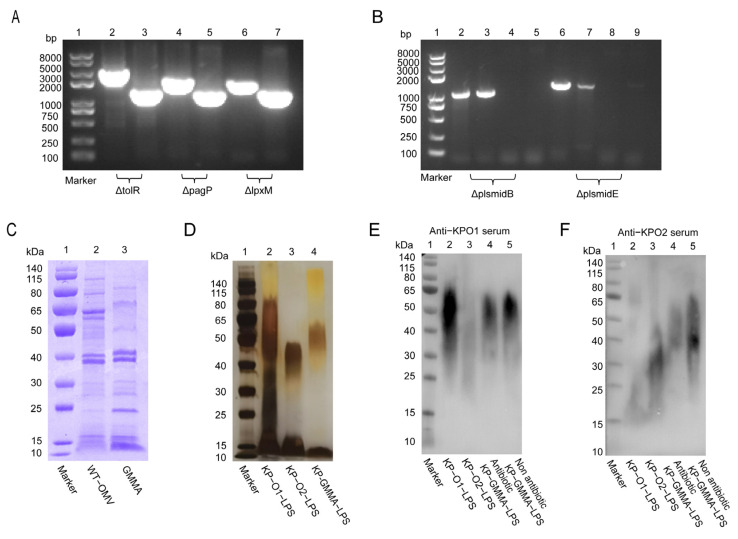
Generation of *K. pneumoniae* gene-mutated strains and co-expression of O1 and O2 serotype polysaccharides. (**A**) PCR identification of tolR, pagP, and lpxM gene knockout. Lane 1: DNA marker; lanes 2, 4, 6: PCR amplification result of wild-type strain; lanes 3, 5, 7: PCR amplification result of mutant strain. (**B**) PCR identification of elimination of plasmids B and E. Lane 1: DNA marker; lanes 2, 3, 6, and 7: PCR amplification result of wild-type strain; lanes 4, 5, 8, and 9: PCR amplification result of mutant strain. (**C**) Characterization of protein variations between WT-OMV and GMMAs via Coomassie brilliant blue staining. (**D**) Verification of expression of GMMA polysaccharides through silver staining. (**E**,**F**) Employment of whole-cell antibodies of anti-KP-O11 (*K. pneumonia* O1 serotype polysaccharides) and anti-KP-O2 (*K. pneumonia* O2 serotype polysaccharides) to ascertain expression levels of polysaccharides within GMMAs both in presence and absence of antibiotics.

**Figure 2 vaccines-13-00226-f002:**
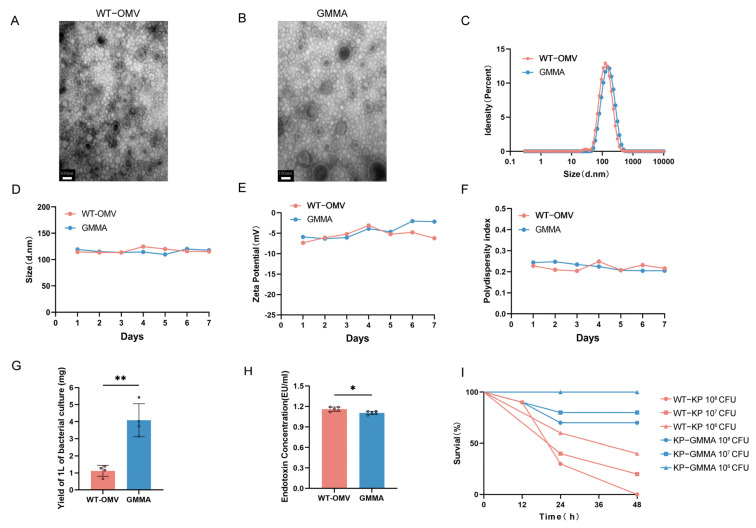
Characterization of WT-OMV and GMMA. (**A**,**B**) TEM images of bacterial cells producing WT-OMV and GMMA. (**C**) DLS assay of WT-OMV and GMMA nanoparticles. (**D**–**F**) Stability of WT-OMV and GMMA at room temperature was detected, n = 3. (**G**) Detection of production of OMVs. (**H**) Detection of content of LPS in OMVs; n = 5. (**I**) Verification of difference in virulence before and after mutation; *Galleria mellonella* was used as model, n = 10. Data are presented as mean ± SD, and differences between groups were tested using *t*-test. * *p* < 0.05, ** *p* < 0.01.

**Figure 3 vaccines-13-00226-f003:**
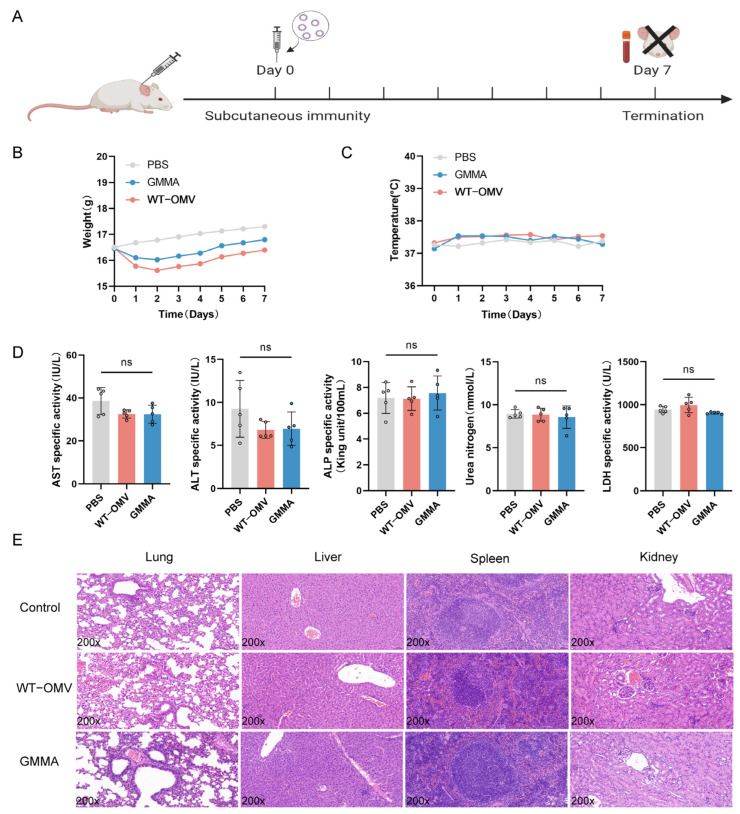
Biocompatibility of GMMA vaccine. (**A**) Single-immunization schematic diagram. (**B**) Single-immunization weight change. (**C**) Single-immunization temperature change. (**D**) Serum biochemical indicators after single immunization; n = 5. (**E**) Pathological changes in tissues of single-immunization mice. Differences between groups were tested using one-way ANOVA. ns *p* > 0.05.

**Figure 4 vaccines-13-00226-f004:**
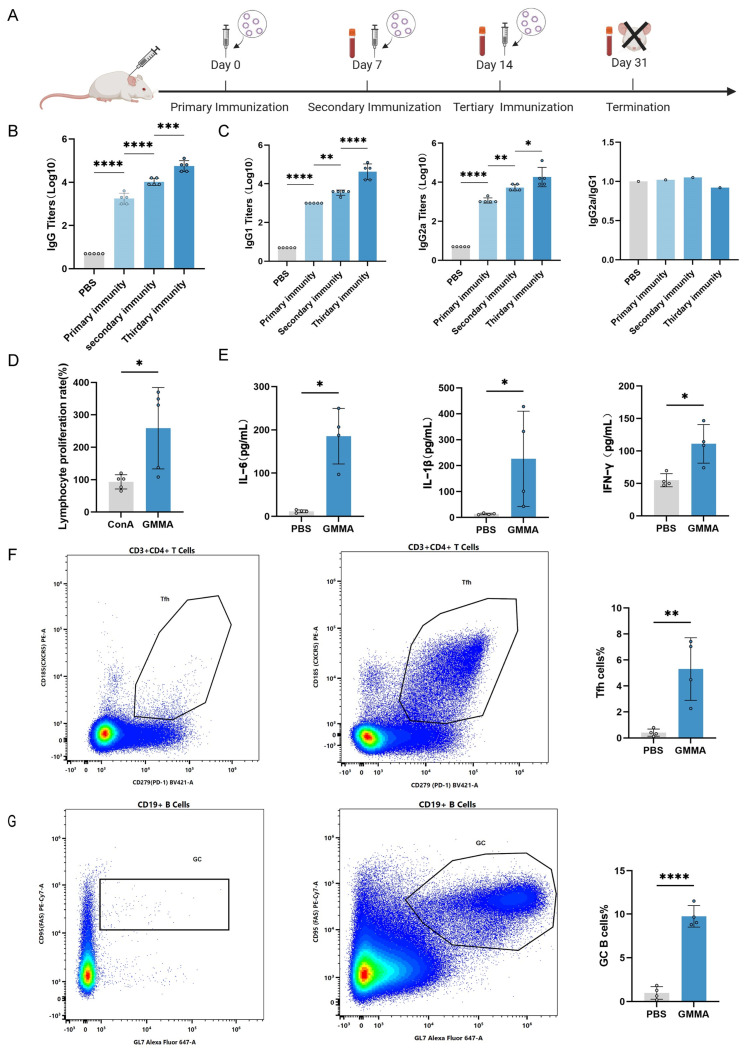
GMMA can trigger bacterial-specific humoral and cellular immune responses. (**A**) GMMA immunization process. (**B**) ELISA detection of serum IgG antibody levels in GMMA immunized mice; n = 5. (**C**) Specific IgG1, IgG2a, and IgG2a/IgG1 ratio in serum after immunization; n = 5. Differences in antibody levels between groups were tested using one-way ANOVA. (**D**) Growth rate of splenic lymphocytes after GMMA stimulation; n = 5. (**E**) GMMAs stimulate splenic lymphocytes to secrete cytokine secretion levels; n = 4. (**F**) Proportion of Tfh cells; n = 4. (**G**) Proportion of GC B cells; n = 4 Different colors represent high-frequency cell subtypes. Differences between two groups were tested using *t*-test. * *p* < 0.05, ** *p* < 0.01, *** *p* < 0.001, **** *p* < 0.0001.

**Figure 5 vaccines-13-00226-f005:**
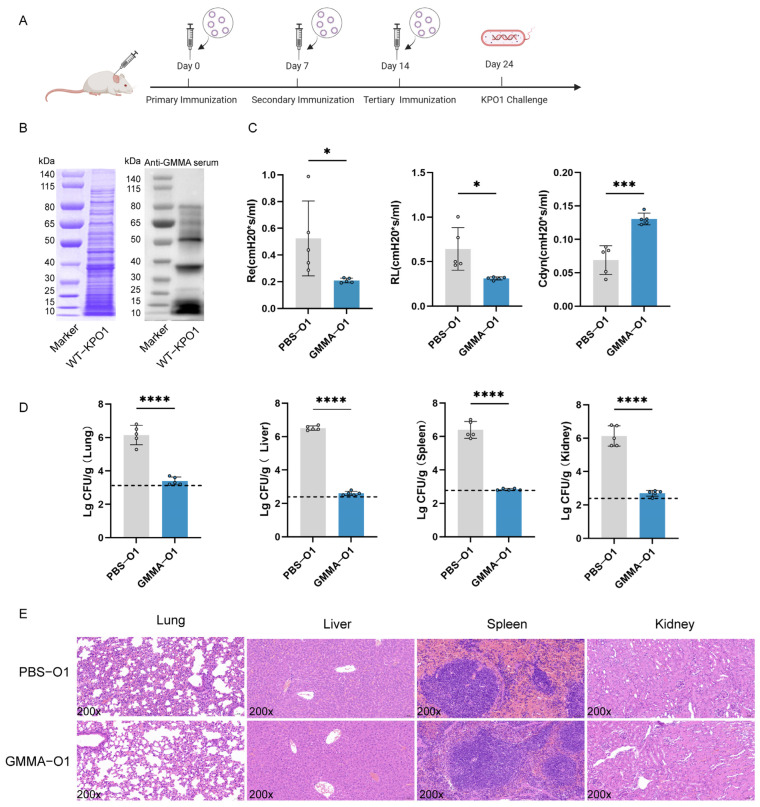
GMMAs triggered immune protection against O1 serotype *K. pneumoniae.* (**A**) Experimental flow chart for protection against O1 serotype *K. pneumoniae* by GMMAs. (**B**) KP-O1 whole bacterial protein was stained with Coomassie brilliant blue, and Western Blot analysis was performed on KP-O1 protein using last immune serum. (**C**) Analysis of lung resistance and lung compliance in mice infected with KP-O1 was conducted using small animal lung function analyzer. (**D**) Bacterial load in mouse organs. Limits of detection (LoD) were shown as dotted lines. (**E**) Analysis of pathological changes in tissues challenged with KP-O1. Data are presented as mean ± SD, and differences between two groups were tested using *t*-test. * *p* < 0.05, *** *p* < 0.001, **** *p* < 0.0001.

**Figure 6 vaccines-13-00226-f006:**
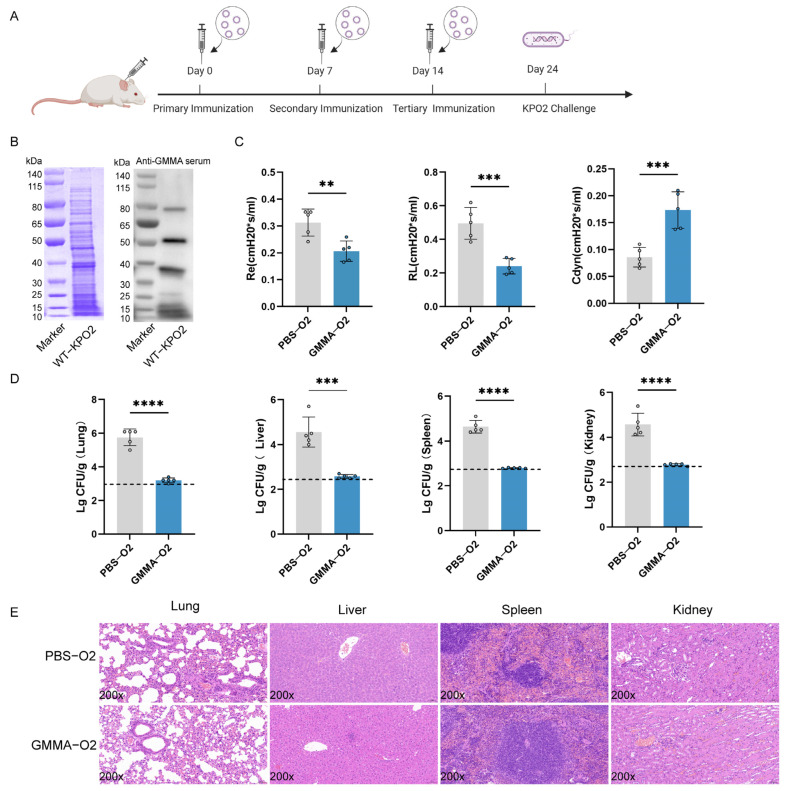
GMMAs triggered immune protection against O2 serotype *K. pneumoniae.* (**A**) Experimental flow chart of protection against for preventing O2 serotype *K. pneumoniae* by GMMAs. (**B**) KP-O2 whole bacterial protein was stained with Coomassie brilliant blue, and Western Blot analysis was performed on KP-O2 protein using the last immune serum. (**C**) Analysis of lung resistance and lung compliance in mice infected with KP-O2 was conducted using small animal lung function analyzer. (**D**) Bacterial load in mouse organs. Limits of detection (LoD) were shown as dotted lines. (**E**) Analysis of pathological changes in tissues of mice challenged with KP-O2. Data are presented as mean ± SD, and differences between the two groups were tested using a *t*-test. ** *p* < 0.01, *** *p* < 0.001, **** *p* < 0.0001.

**Table 1 vaccines-13-00226-t001:** Protective efficacy of GMMA vaccine against O1 and O2 serotypes of *K. pneumoniae*.

Group	PBS-O1	GMMA-O1	PBS-O2	GMMA-O2
RL (cmH20 * s/mL)	0.64268	0.31182	0.49548	0.24208
Re (cmH20 * s/mL)	0.20968	0.20618	0.52474	0.31288
Cdyn (cmH20 * s/mL)	0.1305	0.17324	0.06901	0.085676
Lg CFU/g (Lung)	6.15	3.41	5.70	3.22
Lg CFU/g (Liver)	6.51	2.59	6.5	2.57
Lg CFU/(Spleen)	6.37	2.84	4.64	2.80
Lg CFU/g (Kidney)	6.13	2.70	4.57	2.78

Note: * Data are presented as mean values; *p* < 0.01 indicates that the difference between the data of this group and the corresponding control group is statistically significant.

## Data Availability

Dataset available on request from authors.

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
