# Peer review of "Exploration of a GMMA-Based Bivalent Vaccine Against *Klebsiella pneumoniae"

_vaccines, 2025, doi:10.3390/vaccines13030226_

Round 1

Reviewer 1 Report

Comments and Suggestions for Authors

The manuscript addressed a highly relevant clinical concern: the management of hypervirulent, MDR K. pneumoniae. The work is nicely presented with an introduction that describes that describes this convergence it's importance, and the prevalence of serotypes O1 and O2. It establishes the need for effective vaccines as an alternative to antibiotics, and supports their study design focusing on these two serotypes. As a suggestion, the authors could cite more recent studies on vaccine strategies and include a more comprehensive set of challenges/limitations of the GMMA technology (in the introduction and/or discussion, including references).

Results are presented in a logical and clear manner, aligned with the goal of developing a bivalent vaccine. The conclusions are well supported by the data, particularly regarding the vaccine's immunogenicity, safety, and efficacy against O1 and O2 serotypes of K. pneumoniae. Could you please add statistical values to figure legends 4, 5 and 6? It would also be important to state negative and positive controls for each experiment. The authors should discuss about standardizing vesicle production, as it is an important limitation of this technology, and a possible cross-protection against other serotypes, exploring a broader applicability of their findings. Did the authors consider the longevity of the immune protection, do you have any long-term data on immunity? Although it can be inferred, the authors could more explicitly highlight how each experiment addresses the need for a safe, bivalent vaccine for K. pneumoniae. As a suggestion, a summary table of the results could be helpful for a clear understanding of the results.

The discussion should include a more in depth comparison of this work with recent literature on vaccine development and applications in the context of bacterial infection, and also highlight the contribution of this work to the field.

Comments on the Quality of English Language

The English in this manuscript is generally fine. In order to improve readability the authors should address a few grammar issues such as verb tenses, word order, misplaced words,  and plural/singular, and repeated words and sentences which are too long.

Author Response

Reply to Reviewer 1

The manuscript addressed a highly relevant clinical concern: the management of hypervirulent, MDR K. pneumoniae. The work is nicely presented with an introduction that describes that describes this convergence it's importance, and the prevalence of serotypes O1 and O2. It establishes the need for effective vaccines as an alternative to antibiotics, and supports their study design focusing on these two serotypes. As a suggestion, the authors could cite more recent studies on vaccine strategies and include a more comprehensive set of challenges/limitations of the GMMA technology (in the introduction and/or discussion, including references).

Reply: We thank the Reviewer for recognizing the interest of the topic, and for sharing their views on how we might further strengthen our work. We have added the updated studies about GMMA-based vaccines in the instruction of the revised manuscript.

Results are presented in a logical and clear manner, aligned with the goal of developing a bivalent vaccine. The conclusions are well supported by the data, particularly regarding the vaccine's immunogenicity, safety, and efficacy against O1 and O2 serotypes of K. pneumoniae. Could you please add statistical values to figure legends 4, 5 and 6? It would also be important to state negative and positive controls for each experiment.

Reply: Thank you for your valuable and detailed feedback. We apologize for the statistical values missing from Figs 4, 5, and 6 in the legends. We have addressed the points you raised, and the explanations are provided in the figure legends.

The authors should discuss about standardizing vesicle production, as it is an important limitation of this technology, and a possible cross-protection against other serotypes, exploring a broader applicability of their findings.

Reply: We are grateful for the reviewer’s suggestions. The analysis of GMMA yield indicated that mutant strains produced approximately three times the amount of GMMA compared to wild strains (Fig 2G). This suggests that genetic modification significantly enhances GMMA yield. Approximately 3 mg of GMMA can be harvested from 1L culture. Although a detailed investigation into GMMA yield was not conducted, our findings confirm that mutation enhances the strain's ability to produce GMMA. Future research could explore this issue in greater depth. Based on our current knowledge of other strains, this level of GMMA yield generally remains at a comparable level.

In the present study, we exclusively investigated the immune protection conferred by the vaccine against K. pneumoniae of O1 and O2 serotypes. However, we did not extend our research to include other serotypes of K. pneumoniae. Future research should consider incorporating virus challenge tests using KP of additional serotypes to evaluate and confirm the vaccine's protective efficacy against these other serotypes.

 Did the authors consider the longevity of the immune protection, do you have any long-term data on immunity? Although it can be inferred, the authors could more explicitly highlight how each experiment addresses the need for a safe, bivalent vaccine for K. pneumoniae.

Reply: Thank you for your valuable suggestions. In the present study, we have not yet evaluated the long-term immune efficacy of the vaccine, which represents a limitation of this study. However, in subsequent research, we will assess the immune protection efficacy of the vaccine and other important characteristics to provide experimental evidence for the further development of this vaccine candidate.

As a suggestion, a summary table of the results could be helpful for a clear understanding of the results.

Reply: Thank you for your valuable suggestions. A summary table has been added to the reviewed manuscript.

The discussion should include a more in depth comparison of this work with recent literature on vaccine development and applications in the context of bacterial infection, and also highlight the contribution of this work to the field.

Reply: Thank you for your valuable suggestions. In our revised manuscript, we have added relevant discussions, particularly comparing the differences between our GMMA vaccine and the natural OMV vaccine. This highlights the advantages of the vaccine developed in this study more effectively.

Reviewer 2 Report

Comments and Suggestions for Authors

This study focuses on developing a GMMA-based vaccine using Klebsiella pneumoniae strains. By sequentially knocking out three genes and removing resistance plasmids, a chassis strain (KP-GMMA) was engineered to produce heterozygous O1/O2 lipopolysaccharides (LPS). Compared to wild-type strains, KP-GMMA showed reduced toxicity, stable OMV production, and enhanced immune properties. Immunization trials in mice demonstrated robust humoral and cellular responses, with balanced Th1/Th2 activation and increased Tfh and GC B cells. The vaccine effectively protected mice from infections caused by O1 and O2 serotypes, reducing bacterial loads and mitigating lung inflammation. These findings highlight GMMA's potential as a safe and effective vaccine candidate. However, several changes and clarifications are required throughout the manuscript.

In general terms, it is recommended that the Materials and Methods, as well as the Results sections, be rewritten in a more structured format, providing relevant information that is currently missing from the manuscript. Furthermore, the Discussion section does not thoroughly analyze the obtained results; instead, it merely summarizes them without comparing them to those previously reported by other authors. It also fails to highlight the improvements or novel contributions that this work offers in relation to prior research.

General Comments:

-          What statistical analyses have been conducted in the different sections? This information should be included, either in a dedicated section or within the methodology section describing each experiment.

-          In the animal models’ section, how many mice were used for each test? The sample size (n) is only specified for one test, where n = 5. This information should be provided for all tests to ensure a proper evaluation of the manuscript.

-          The nomenclature of strains and genes should be reviewed throughout the document, as they are not consistently formatted (e.g., use of italics) in several sections.

-          All images should include more detailed legends that clearly explain the content and indicate the meaning of any abbreviations used. Additionally, the images should be clearly annotated, and reference should be made to the statistical tests used if applicable.

Specific comments:

-          Lines 85-86: The strain names in the text do not match those in Table S1. Different nomenclatures are used throughout the document to refer to the same strains. It is essential to standardize a single naming convention and maintain consistency throughout the manuscript.

-          Lines 93-94: Strain KP-02 has been sequenced; however, the Accession Number is not provided anywhere in the document for sequence reference. This information should be included either in a separate section or within the Results section (Line 217).

-          Line 98-99: What was the rationale for selecting these three genes (rpoB, pagP and lpxM? Were previous trials conducted, or was the decision based on a review of the existing literature?

-          Line 117: It should be specified that the primers used are listed in Table S2.

-          Line 166-176: In line with the general comment regarding the need to clarify the Materials and Methods and Results sections, as well as to specify the number of mice used in each assay, the number of mice per group and the type of inoculation used in each assay should be clearly stated here. It is currently unclear which assay corresponds to which experimental condition.

-          Line 178: What type of administration was used? Intranasal?

-          Line 184: What immunization schedule was followed? How were spleen cell suspensions prepared?

-          Line 197: What sera were used for the ELISA assay?

-          Line 199: What secondary antibody was used in this assay?

-          Line 204: The immunization schedule and the number of mice per group should be provided.

-          Line 211: Correct the nomenclature of K. pneumoniae.

-          Line 213: Correct the nomenclature of the KP-02 strain.

-          Line 219: Indicate the table that contains the primer sequences.

-          Line 216-223: Some of the information provided in this paragraph should be included in the Materials and Methods section.

-          Line 227: Why is this effect attributed to the inactivation of the tolR gene?

-          Line 229: Streptococcus pneumoniae is named instead of Klebsiella pneumoniae.

-          Line 264: It is stated that 10 proteins from each type of vesicle (WT and GMMA) are listed in Table S3; however, only 9 proteins are included for the GMMA vesicles. The missing protein should be added, or the number of listed proteins should be corrected accordingly.

-          Line 269-270: Check and correct spacing issues.

-          Figure S1: Indicate which figure corresponds to each dose. Additionally, include a figure comparing the IgG titers obtained after each immunization with the different OMV concentrations for a more visual representation. Once the optimal OMV concentration has been determined, justify the selection in the text, explaining the criteria used for the decision.

-          Line 279-287: This trial, which tests the safety of the vaccine, is not mentioned in the Materials and Methods section. It should be included there.

-          Figure 4: The immunization schedule only shows the serum collection after the last vaccine dose; however, antibody titers are measured after each dose. If an immunization schedule is provided, all procedures carried out should be detailed.

-          Figure 4B: The legend does not explain the meaning of the abbreviations used. Additionally, each graph should be individually identified (this applies to all other panels and figures as well).

-          Line 305: Figure 4E does not correspond to the result described in the text.

-          Line 314-325: This information is repetitive and should be revised.

-          Line 367: How was the bacterial load determined? This method is not detailed in the Materials and Methods section.

-          Reference number 24 does not follow the bibliographic style of the other references.

Author Response

Reply to Reviewer 2

This study focuses on developing a GMMA-based vaccine using Klebsiella pneumoniae strains. By sequentially knocking out three genes and removing resistance plasmids, a chassis strain (KP-GMMA) was engineered to produce heterozygous O1/O2 lipopolysaccharides (LPS). Compared to wild-type strains, KP-GMMA showed reduced toxicity, stable OMV production, and enhanced immune properties. Immunization trials in mice demonstrated robust humoral and cellular responses, with balanced Th1/Th2 activation and increased Tfh and GC B cells. The vaccine effectively protected mice from infections caused by O1 and O2 serotypes, reducing bacterial loads and mitigating lung inflammation. These findings highlight GMMA's potential as a safe and effective vaccine candidate. However, several changes and clarifications are required throughout the manuscript.

Reply: We thank the Reviewer for recognizing the interest of the topic, and for sharing their views on how we might further strengthen our work.

In general terms, it is recommended that the Materials and Methods, as well as the Results sections, be rewritten in a more structured format, providing relevant information that is currently missing from the manuscript. Furthermore, the Discussion section does not thoroughly analyze the obtained results; instead, it merely summarizes them without comparing them to those previously reported by other authors. It also fails to highlight the improvements or novel contributions that this work offers in relation to prior research.

Reply: Thank you for your valuable and detailed feedback. We have carefully revised the methods and results sections, providing additional details that were not previously mentioned to enhance the readability of the article. We have also modified the results section to present our research findings more accurately. Detailed changes are clearly marked in the manuscript. In our revised manuscript, we also have added relevant discussions, particularly comparing the differences between our GMMA vaccine and the natural OMV vaccine. This highlights the advantages of the vaccine developed in this study more effectively.

General Comments:

What statistical analyses have been conducted in the different sections? This information should be included, either in a dedicated section or within the methodology section describing each experiment.

Reply: Thank you for your valuable feedback. We apologize for missing the description of the statistical method. We have addressed the points you raised to improve the clarity and readability of our manuscript.

In the animal models’ section, how many mice were used for each test? The sample size (n) is only specified for one test, where n = 5. This information should be provided for all tests to ensure a proper evaluation of the manuscript.

Reply: Thank you for your valuable feedback. We have provided the quantity information for all animal tests.

The nomenclature of strains and genes should be reviewed throughout the document, as they are not consistently formatted (e.g., use of italics) in several sections.

Reply: Thank you for your valuable and detailed feedback. We have corrected the relevant content in the revised version of the manuscript.

All images should include more detailed legends that clearly explain the content and indicate the meaning of any abbreviations used. Additionally, the images should be clearly annotated, and reference should be made to the statistical tests used if applicable.

Reply: Thank you for your valuable and detailed feedback. We have addressed the points you raised to improve the clarity, readability, and robustness of our manuscript.

Specific comments:

-          Lines 85-86: The strain names in the text do not match those in Table S1. Different nomenclatures are used throughout the document to refer to the same strains. It is essential to standardize a single naming convention and maintain consistency throughout the manuscript.

Reply: Thank you for your valuable and detailed feedback. We have revised this point as required in the revised manuscript.

          Lines 93-94: Strain KP-02 has been sequenced; however, the Accession Number is not provided anywhere in the document for sequence reference. This information should be included either in a separate section or within the Results section (Line 217).

Reply: Thank you for your valuable suggestions.  We submitted the full sequence of this strain to NCBI when the manuscript was reviewed. The Accession Numbers have been provided in the revised manuscript.

 -          Line 98-99: What was the rationale for selecting these three genes (rpoB, pagP and lpxM? Were previous trials conducted, or was the decision based on a review of the existing literature?

Reply: Thank you for your valuable feedback. We have added content explaining our reasons for choosing these target genes in the revised manuscript.

-          Line 117: It should be specified that the primers used are listed in Table S2.

Reply: Thank you for your valuable suggestions. We have added relevant descriptions in the revised manuscript.

-          Line 166-176: In line with the general comment regarding the need to clarify the Materials and Methods and Results sections, as well as to specify the number of mice used in each assay, the number of mice per group, and the type of inoculation used in each assay should be clearly stated here. It is currently unclear which assay corresponds to which experimental condition.

Reply: Thank you for your valuable suggestions. We apologize for the confusion we caused due to our unclear description. We have added relevant descriptions in the revised manuscript.

-          Line 178: What was used? Intranasal?

Reply: Thank you for your valuable feedback. We apologize for missing the description of the administration type. Unless otherwise specified, the immunization method is uniformly administered via subcutaneous injection. We have addressed this information in the revised manuscript.

-          Line 184: What immunization schedule was followed? How were spleen cell suspensions prepared?

Reply: Thank you for your valuable feedback. We have addressed this information in the revised manuscript.

-          Line 197: What sera were used for the ELISA assay?

Reply: Thank you for your valuable feedback. The information on sera testing has been introduced in the revised manuscript.

-          Line 199: What secondary antibody was used in this assay?

Reply: Thank you for your valuable feedback. We have addressed the information on secondary antibodies in the revised manuscript.

-          Line 204: The immunization schedule and the number of mice per group should be provided.

Reply: Thank you for your valuable feedback. We have provided the quantity information and immunization schedule for all animal tests.

-          Line 211: Correct the nomenclature of K. pneumoniae.

Reply: Thank you for your valuable feedback. We have corrected this point in the revised manuscript.

-          Line 213: Correct the nomenclature of the KP-02 strain.

Reply: Thank you for your valuable feedback. We have corrected this point in the revised manuscript.

-          Line 219: Indicate the table that contains the primer sequences.

Reply: Thank you for your valuable feedback. We have corrected this point in the revised manuscript.

-          Line 216-223: Some of the information provided in this paragraph should be included in the Materials and Methods section.

Reply: Thank you for your valuable suggestions. We have rewritten this section according to your comments in the revised manuscript.

-          Line 227: Why is this effect attributed to the inactivation of the tolR gene?

Reply: Thank you for your valuable feedback. Upon careful consideration, we agree that our initial interpretation may have been premature. While it is true that tolR gene knockout alters cell membrane stability, whereas lpxM and pagP only modify LPS structure, we acknowledge that we lack sufficient evidence to conclusively attribute the observed protein band changes solely to the tolR knockout. Therefore, we have removed this statement from the main text as suggested.

-          Line 229: Streptococcus pneumoniae is named instead of Klebsiella pneumoniae.

Reply: Thank you for your valuable feedback. We apologize for the careless mistake in our manuscript. We have corrected it in the revised manuscript.

-          Line 264: It is stated that 10 proteins from each type of vesicle (WT and GMMA) are listed in Table S3; however, only 9 proteins are included for the GMMA vesicles. The missing protein should be added, or the number of listed proteins should be corrected accordingly.

Reply: Thank you for your valuable feedback. We have corrected this mistake in the revised manuscript.

-          Line 269-270: Check and correct spacing issues.

Reply: Thank you for your valuable feedback. We have corrected this mistake in the revised manuscript.

-          Figure S1: Indicate which figure corresponds to each dose. Additionally, include a figure comparing the IgG titers obtained after each immunization with the different OMV concentrations for a more visual representation. Once the optimal OMV concentration has been determined, justify the selection in the text, explaining the criteria used for the decision.

Reply: Thank you for your valuable suggestions. We have revised the y-axis of Figure S1 to differentiate the antibody titers produced by three immunizations at different GMMA immunization doses.

-          Line 279-287: This trial, which tests the safety of the vaccine, is not mentioned in the Materials and Methods section. It should be included there.

Reply: Thank you for your valuable suggestions. We have rewritten this section according to your comments in the revised manuscript.

-        Figure 4: The immunization schedule only shows the serum collection after the last vaccine dose; however, antibody titers are measured after each dose. If an immunization schedule is provided, all procedures carried out should be detailed.

Reply: Thank you for your valuable suggestions. We apologize for the confusion we caused due to our unclear description. We have changed the relevant figure in the revised manuscript.

-          Figure 4B: The legend does not explain the meaning of the abbreviations used. Additionally, each graph should be individually identified (this applies to all other panels and figures as well).

Reply: Thank you for your valuable suggestions. We apologize for the confusion we caused due to our unclear description. Detailed explanations for the abbreviations have been provided in the new manuscript. At the same time, we also made similar changes to other images.

-          Line 305: Figure 4E does not correspond to the result described in the text.

Reply: Thank you for your valuable feedback. We have rechecked and added Figure 4 E in the article, which is consistent with the results shown in the image.

-        Line 314-325: This information is repetitive and should be revised.

Reply: Thank you for your valuable feedback. We have corrected this mistake in the revised manuscript.

-          Line 367: How was the bacterial load determined? This method is not detailed in the Materials and Methods section.

Reply: Thank you for your valuable suggestions. We apologize for the confusion we caused due to our unclear description in the method section, we have restated the method for determining bacterial load.

-          Reference number 24 does not follow the bibliographic style of the other references.

Reply: Thank you for your valuable feedback. We have corrected this mistake in the revised manuscript.

Reviewer 3 Report

Comments and Suggestions for Authors

The presented original article by Qikun Ou et al. „ Exploration of a GMMA-based bivalent vaccine against Klebsiella pneumoniae “ is a well-conducted and written research with an aim to develop a GMMA-based bivalent vaccine for O1 and O2 types of Klebsiella pneumoniae. Congratulations to the authors on well design of the study.

A few minor remarks or suggestions for the authors.

Generally, after the first introduction of the abbreviation, use it and do not reintroduce it in every paragraph. Check throughout the text.

Abstract:

Line 26 - K.pneumoniae – italics, check throughout the text (line 349, 353 )

Introduction:

Line 38 – define ”the other group”  - hvKp, what are its characteristics

Line 41 – provide at least one additional sentence with major characteristics of fusion phenomena of cKp and hvKp

Line 50 – First mention of LPS – introduce the abbreviation and erase it from line 65

Line 52 – 8 change in eight

Line 58 – bivalent

Line 59 – several types of vaccines (in research – none is yet licensed)

Line 70 - outer membrane vesicles  - OMV (use the abbreviation, Line 262, 397, 410, 421, 446)

Line 71-72 - Add a sentence regarding the knockout of three specific genes tolR,               pagP, and lpxM, and why you choose them for a knockout

Line 73 – Missing the aim of the research. In this study, we aimed to…. Afterward, you may in fewer sentences give in brief lines 73-81

Materials and Methods:

Line 90 - a single colony was inoculated into 5 mL

Line 91 - shake at 220 rpm and 37°C for 8-9 hours to reach the plateau period before applying it to subsequent experiments.

Line 93 – Add another brief section regarding the sequences of the genome and plasmids on Illumina sequencing.

Line 110 – PCR ( introduce the abbreviation, delete from Line 214)

Line 123 – A 100 μL of…

Line   125 – 15 min.

Line 129 - By spreading aliquots on agar plates to detect bacterial growth. – Something is missing

Line 135 - with Coomassie (brilliant – missing)

Line 152, 154, 250 - Galleria mellonella (Italic)

Results:

Line 211 – pneumonia (small P, also in Table S1)

Line 220O1 and O2

Line 229 - Streptococcus pneumonia – Klebsiella pneumoniae

Line 239 - the co-expression of O1 and O2

Line  245 - via Coomassie brilliant blue

Line 251 – GMMA – introduced previously in Line 72, 409, 442,451

Line 254 – DLS – introduced previously in Line 133

 Line 258 – LPS – introduced previously in Line 50, 414, 434,438

Discussion:

The general impression is that despite numerous results, there is a lack of discussion.

Line 392 – MDR - introduced previously in Line 42

Line 393 – Enterobacteriaceae – italic

Line 399 – Toll-like

Line 401 – PAMP – introduced previously in Line 64

The conclusion should be rewritten based on the use of lines 72-81.

Author Response

Reply to Reviewer 3

The presented original article by Qikun Ou et al. „ Exploration of a GMMA-based bivalent vaccine against Klebsiella pneumoniae “ is a well-conducted and written research with an aim to develop a GMMA-based bivalent vaccine for O1 and O2 types of Klebsiella pneumoniae. Congratulations to the authors on well design of the study.

Reply: We thank the Reviewer for recognizing the interest of the topic, and for sharing their views on how we might further strengthen our work.

A few minor remarks or suggestions for the authors.

Generally, after the first introduction of the abbreviation, use it and do not reintroduce it in every paragraph. Check throughout the text.

Abstract:

Line 26 - K.pneumoniae – italics, check throughout the text (line 349, 353 )

Introduction:

Reply: Thank you for your valuable feedback. We have corrected this mistake in the revised manuscript.

Line 38 – define ”the other group”  - hvKp, what are its characteristics

Reply: Thank you for your valuable feedback. We have provided the explanations for hvKp in the new manuscript.

Line 41 – provide at least one additional sentence with major characteristics of fusion phenomena of cKp and hvKp

Reply: Thank you for your valuable suggestions. We have added the content to explain this point in the revised manuscript.

Line 50 – First mention of LPS – introduce the abbreviation and erase it from line 65

Reply: Thank you for your valuable feedback. We have corrected this point in the revised manuscript.

Line 52 – 8 change in eight

Reply: Thank you for your valuable feedback. We have changed it in the revised manuscript.

Line 58 – bivalent

Reply: Thank you for your valuable feedback. We have corrected it in the revised manuscript.

Line 59 – several types of vaccines (in research – none is yet licensed)

Reply: Thank you for your valuable feedback. We have rewritten this part in the revised manuscript.

Line 70 - outer membrane vesicles  - OMV (use the abbreviation, Line 262, 397, 410, 421, 446)

Reply: Thank you for your valuable feedback. We have corrected it in the revised manuscript.

Line 71-72 - Add a sentence regarding the knockout of three specific genes tolR, pagP, and lpxM, and why you choose them for a knockout

Reply: Thank you for your valuable suggestions.  We have added content explaining our reasons for choosing this target genes in the method section of the revised manuscript.

Line 73 – Missing the aim of the research. In this study, we aimed to…. Afterward, you may in fewer sentences give in brief lines 73-81

Reply: Thank you for your valuable suggestions. We have rewritten this section in the revised manuscript.

Materials and Methods:

Line 90 - a single colony was inoculated into 5 mL

Reply: Thank you for your valuable suggestions. We have rewritten this sentence in the revised manuscript.

Line 91 - shake at 220 rpm and 37°C for 8-9 hours to reach the plateau period before applying it to subsequent experiments.

Line 93 – Add another brief section regarding the sequences of the genome and plasmids on Illumina sequencing.

Reply: Thank you for your valuable feedback. We have added this content in the revised manuscript.

Line 110 – PCR ( introduce the abbreviation, delete from Line 214)

Line 123 – A 100 μL of…

Reply: Thank you for your valuable feedback. We have changed it in the revised manuscript.

Line   125 – 15 min.

Reply: Thank you for your valuable feedback. We have changed it in the revised manuscript.

Line 129 - By spreading aliquots on agar plates to detect bacterial growth. – Something is missing

Reply: Thank you for your valuable feedback. We have changed it in the revised manuscript.

Line 135 - with Coomassie (brilliant – missing)

Reply: Thank you for your valuable feedback. We have changed it in the revised manuscript.

Line 152, 154, 250 - Galleria mellonella (Italic)

Reply: Thank you for your valuable feedback. We have changed it in the revised manuscript.

Results:

Line 211 – pneumonia (small P, also in Table S1)

Reply: Thank you for your valuable feedback. We have changed it in the revised manuscript.

Line 220 – O1 and O2

Reply: Thank you for your valuable feedback. We have changed it in the revised manuscript.

Line 229 - Streptococcus pneumonia – Klebsiella pneumoniae

Reply: Thank you for your valuable feedback. We apologize for the careless mistake in our manuscript. We have corrected it in the revised manuscript.

Line 239 - the co-expression of O1 and O2

Reply: Thank you for your valuable feedback. We apologize for the careless mistake in our manuscript. We have corrected it in the revised manuscript.

Line  245 - via Coomassie brilliant blue

Reply: Thank you for your valuable feedback. We have changed it in the revised manuscript.

Line 251 – GMMA – introduced previously in Line 72, 409, 442,451

Reply: Thank you for your valuable feedback. We have changed it in the revised manuscript.

Line 254 – DLS – introduced previously in Line 133

Reply: Thank you for your valuable feedback. We have changed it in the revised manuscript.

 Line 258 – LPS – introduced previously in Line 50, 414, 434,438

 Reply: Thank you for your valuable feedback. We have changed it in the revised manuscript.

Discussion:

The general impression is that despite numerous results, there is a lack of discussion.

Line 392 – MDR - introduced previously in Line 42

Reply: Thank you for your valuable feedback. We have changed it in the revised manuscript.

Line 393 – Enterobacteriaceae – italic

Reply: Thank you for your valuable feedback. We have changed it in the revised manuscript.

Line 399 – Toll-like

Reply: Thank you for your valuable feedback. We have changed it in the revised manuscript.

Line 401 – PAMP – introduced previously in Line 64

  Reply: Thank you for your valuable feedback. We have changed it in the revised manuscript.

The conclusion should be rewritten based on the use of lines 72-81.

Reply: Thank you for your valuable suggestions. We have rewritten this section in the revised manuscript.